# Epidemiologic and Clinical Characteristics of Human Bocavirus Infection in Children with or without Acute Gastroenteritis in Acre, Northern Brazil

**DOI:** 10.3390/v15041024

**Published:** 2023-04-21

**Authors:** Fábyla D’ Tácia Brito Trindade, Endrya Socorro Foro Ramos, Patrícia Santos Lobo, Jedson Ferreira Cardoso, Edvaldo Tavares Penha Júnior, Delana Andreza Melo Bezerra, Mayara Annanda Oliveira Neves, Jorge Alberto Azevedo Andrade, Monica Cristina Moraes Silva, Joana D’Arc Pereira Mascarenhas, Sylvia Fátima Santos Guerra, Luana Silva Soares

**Affiliations:** 1Postgraduate Program in Epidemiology and Health Surveillance, Instituto Evandro Chagas, Secretaria de Vigilância em Saúde e Ambiente—IEC/SVSA/MS, Ananindeua 67030-000, PA, Brazil; 2Seção de Virologia, Instituto Evandro Chagas, Secretaria de Vigilância em Saúde e Ambiente—IEC/SVSA/MS, Ananindeua 67030-000, PA, Brazil; 3Secretaria de Estado da Saúde do Pará—SESPA, Belém 66093-677, PA, Brazil

**Keywords:** human bocaparvovirus, acute gastroenteritis, children

## Abstract

Human bocavirus (HBoV) is an emerging virus detected around the world that may be associated with cases of acute gastroenteritis (AGE). However, its contribution to AGE has not been elucidated. This study aimed to describe the frequency, clinical features, and HBoV species circulation in children up to 5 years with or without AGE symptoms in Acre, Northern Brazil. A total of 480 stool samples were collected between January and December 2012. Fecal samples were used for extraction, nested PCR amplification, and sequencing for genotyping. Statistical analysis was applied to verify the association between epidemiological and clinical characteristics. Overall, HBoV-positivity was 10% (48/480), with HBoV-positive rates of 8.4% (19/226) and 11.4% (29/254) recorded in diarrheic and non-diarrheic children, respectively. The most affected children were in the age group ranging between 7 and 24 months (50%). HBoV infection was more frequent in children who live in urban areas (85.4%), use water from public networks (56.2%), and live with adequate sewage facilities (50%). Co-detection with other enteric viruses was 16.7% (8/48) and the most prevalent coinfection was RVA+ HBoV (50%, 4/8). HBoV-1 was the most frequent species detected in diarrheic and non-diarrheic children, responsible for 43.8% (21/48) of cases, followed by HBoV-3 (29.2%, 14/48) and HBoV-2 (25%, 12/48). In this study, HBoV infection was not always associated with AGE, as most HBoV cases belonged to the non-diarrheal group. Future studies are warranted in order to determine the role of HBoV in causing acute diarrhea disease.

## 1. Introduction

Acute gastroenteritis (AGE) is still a leading cause of childhood morbidity and mortality worldwide, accounting for nearly 450,000 deaths of children < 5 years of age in 2019, mostly in low-income countries [1]. Enteric viruses are the most frequent common pathogens causing AGE, responsible for approximately 50% of cases. Rotavirus A (RVA), norovirus (NoV), human adenovirus (HAdV), sapovirus (SaV), and astrovirus (AstV) are the most common viral agents worldwide [2]. However, in a significant number of AGE cases (up to 50%), the causative agent can remain undiagnosed [3,4].

Human bocavirus (HBoV), which belongs to the *Parvoviridae* family, as part of the *Bocaparvovirus* genus, was first discovered in 2005 in children with acute respiratory tract symptoms [5,6]. Later, HBoV was identified in human stool samples from children with AGE [7,8,9]. HBoV is currently considered to be an emerging virus and it is still unclear whether it could be associated with cases of AGE due to the high rate of the co-detection pattern of HBoV with other gastroenteric viruses in symptomatic patients, as well as its frequent detection in asymptomatic individuals [10,11].

HBoV is a small non-enveloped icosahedral virus (20 nm) with a single-stranded DNA genome containing about 5.3 kilobases. Its genome is organized into three open reading frames (ORFs) that encode for two nonstructural proteins (NS1 and NP1) and two viral capsid proteins (VP1 and VP2) [11,12]. HBoV is divided into four species (HBoV-1 to HBoV-4), based on the genomic analysis of VP1 and VP2 proteins [5,7,13]. HBoV-1 is frequently associated with respiratory tract infections; however, it has been detected in AGE cases. On the other hand, the other species are mainly found in stool samples of children and adults with and without gastrointestinal symptoms [10,14,15].

Therefore, this study aimed to investigate the frequency of HBoV infection in children up to 5 years with or without AGE symptoms in Acre, Northern Brazil. In order, we screened the positive HBoV samples for other major gastroenteric viruses, such as RVA, NoV, and HAstV, in order to determine whether the causative agent of diarrhea was in the collected samples. Additionally, we investigated the correlation between HBoV detection and epidemiological, demographic, economic, and clinical features.

## 2. Materials and Methods

### 2.1. Clinical Specimens and Ethical Aspects

A cross-sectional retrospective study was carried out in Rio Branco, Acre, between January and December 2012. A total of 480 stool samples were collected from children aged < 5 years with (226 children) or without (254 children) AGE symptoms, and were admitted to a local hospital (collections made up to 48 h from admission) or subject to outpatient clinic treatment. Standard questionnaires were filled out with epidemiological and clinical information. Fecal samples had already been previously tested for RVA, NoV, and HAstV [16,17,18]. This study was approved by the Evandro Chagas Institute Ethical Research Committee (protocol number 3.383.249), in accordance with National Health Council’s Resolution 466/2012.

### 2.2. Nucleic Acid Extraction

Viral nucleic acids (DNA and RNA viruses) were extracted from 10% fecal suspensions with Tris-calcium buffer (pH = 7.2) using silica glass powder [19]. The isolated nucleic acid was kept frozen at −70 °C until the molecular analysis was carried out. In each extraction procedure, RNAse/DNAse-free water was used as a negative control.

### 2.3. HBoV Molecular Detection

HBoV detection was performed by a PCR, followed by a nested PCR, using two sets of primers that targeted a variable VP1/VP2 region, as described previously [13]. The PCR products were detected by electrophoresis on 1.5% agarose gel. The presence of HBoV was determined through a specific-sized amplicon corresponding to the second round of nested PCR of 576 bp, after being stained with a nucleic acid staining solution and visualized with a UV transilluminator.

### 2.4. Molecular Characterization and Phylogenetic Analysis

The amplicons obtained were purified using ExoSAP-IT^®^ enzymes (Applied Biosystems), according to the manufacturer’s recommendations. The purified PCR products were then subjected to direct sequencing using the same primers of nested PCR. The sequencing reaction was performed with a commercial BigDye Terminator v.3.1 Cycle Sequencing Kit (Applied Biosystems, CA, USA) in an ABI Prism 3130xl automated sequencer (Applied Biosystems, Foster City, CA, USA).

The sequences obtained following sequencing analysis were assembled by de novo and reference mapping methods, and were then edited and curated using Geneious software (v.7) [20]. The sequence alignment, together with other related sequences available on GenBank (www.ncbi.nlm.nih.gov), was performed using the MAFFT algorithm [21]. To construct the phylogenetic tree of the HBoV VP1/VP2 partial genes, they were constructed with the maximum likelihood method using FastTree v.2.1.11 software, including the GTR+Gamma+F nucleotide substitution model. Bootstrap values of the nodes indicate support of 1000 replicas, obtaining reproducible results and awarding greater reliability to the clusters. Nucleotide distances were generated from an identity matrix in MEGA v.X software. Partial nucleotide sequences from this study were deposited in the GenBank database (http://www.ncbi.nlm.nih.gov) under the access numbers OQ695608-OQ695654.

### 2.5. Statistical Analysis

All statistical analyses were performed in a database created by the Statistical Package Software for Social Sciences (SPSS) (version 20.0). The baseline data were submitted for descriptive analysis to obtain the prevalence of the investigated outcomes. Bivariate analysis was carried out to verify the association between independent variables and HBoV infection using the chi-square test (x2). A *p*-value ≤ 0.05 was considered statistically significant.

## 3. Results

Overall, HBoV positivity was 10% (48/480), with HBoV-positive rates of 8.4% (19/226) and 11.4% (29/254) recorded in diarrheic and non-diarrheic children, respectively. No statistical significance was observed in the analyzed groups.

No correlation was shown between epidemiological aspects and the risk of HBoV infection; however, the most affected children were in the age group ranging between 7 and 24 months, corresponding to 50% of cases. HBoV infection was similar between females (47.9%) and males (52.1%). Regarding epidemiological characteristics, it was reported that HBoV infection was more frequent in children who lived in urban areas (85.4%), used water from the public network (56.2%), and lived with adequate sewage facilities (50%). Concerning family income, 52.1% received up to one monthly minimum wage (Table 1). None of these parameters showed statistical significance.

With respect to clinical features, most children infected with HBoV (60.4%, 29/48) belonged to the non-diarrheic group and sought care for other clinical causes, such as respiratory symptoms and trauma; however, questionnaires were not filled out with detailed symptoms (data not shown). Considering the analyzed symptoms, it was reported that 75.9% (22/29) and 96.6% (28/29) of children in the non-diarrheic group did not present fever and vomiting episodes, respectively, with a statistical difference for these symptoms (*p* ≤ 0.05) (Table 2).

The global prevalence of coinfection involving HBoV and other viruses was 1.7% (8/480). However, when analyzing only HBoV and other viruses with positive specimens, coinfection prevalence was 16.7% (8/48). It was reported that the most prevalent coinfection was RVA+ HBoV (50%, 4/8), followed by NoV+HBoV (37.5%, 3/8) and HAstV+HBoV (12.5%, 1/8). Co-detection with HBoV and at least one virus was shown in diarrheic and non-diarrheic children (Table 3).

Concerning HBoV characterization, 47 positive samples were genotyped. HBoV-1 was the most frequent species detected in diarrheic and non-diarrheic children, responsible for 43.8% (21/48) of cases, followed by HBoV-3 (29.2%, 14/48) and HBoV-2 (25%, 12/48). Based on the phylogenetic inference of the VP1/VP2 gene, HBoV-1 strains grouped with isolates from Brazil, China, Thailand, and European strains that circulated from 2008 to 2018 showed bootstrap support values ranging from 72.6% to 99.2% and had high nucleotide levels (98.6–100%) similarities with contemporaneous strains. HBoV-2 samples were grouped into lineage A and formed two clusters: the first cluster was grouped with Brazillian isolates detected in 2011, 2014, 2016, and 2017 (85.2% bootstrap value), while the second cluster was grouped with strains from Australia (2001), the United Kingdom (2007), and Thailand (2007), with a bootstrap value of 89.3%. Despite the fact that Brazilian HBoV-2 strains were grouped into two clusters, nucleotide similarity ranged from 91 to 99%. HBoV-3 strains were clustered with isolates from Brazil (2003–2016), China (2007–2008), Thailand (2011), Tunisia (2007), and Pakistan (2009), showing bootstrap values ranging from 71.4 to 100%. Similar to Brazilian HBoV-2 strains, HBoV-3 strains formed two clusters and were grouped with nucleotide similarity scores ranging from 89 to 100%. The HBoV-4 type was not identified (Figure 1).

## 4. Discussion

Since HBoV was first identified in 2005, this virus has been found to infect symptomatic and asymptomatic individuals for AGE with varying prevalence levels, including studies carried out in the northern region of Brazil [15,22,23,24]. In the present study, overall HBoV positivity score was 10% and similar rates were detected in diarrheic children (8.4%) and non-diarrheic children (11.4%). Our results corroborate with data from a study conducted by Risku et al. [25] in Finland, where HBoV was detected in 9.2% of children aged 0–15 years old with and without AGE symptoms. However, Risku et al. [25] reported that HBoV was most frequent in diarrheic (9.7%) children as opposed to non-diarrheic (5.4%) children, and no statistical significance was observed in the analyzed groups. A similar frequency was detected by Malta et al. [24] in a study with children up to two years of age with AGE symptoms from ten Brazilian states, where HBoV positivity was 12.4%. Recently, Leitão et al. [26] reported an HBoV detection rate of 14.2% in Amazonian children with AGE symptoms. Otherwise, the global HBoV frequency reported in a study conducted in Russia was 1.1%, where HBoV rates from children with and without AGE symptoms were 1.2% and 0.3%, respectively [27]. The divergence of HBoV frequencies in studies may relate to different methodologies used; stool samples from a collection period; the age of the analyzed population; and the socioeconomic, environmental, and sanitation features of each setting [12,23].

HBoV was mostly detected in children aged 7 to 24 months, corresponding to 50% of cases, followed by those aged 25 to 60 months (39.6%). Previous studies described that children aged ≤ 2 years old were found to be most susceptible to HBoV infection [28,29]. Two studies with hospitalized children presenting AGE symptoms from Brazil reported a higher HBoV positivity rate in children aged ≤ 5 years. Soares et al. [22] investigated 225 fecal samples from children aged less than 10 years with AGE from northern Brazil and reported that 66.6% of HBoV cases affected children aged 7 to 24 months. Furthermore, a study conducted by Malta et al. [24] analyzed stool samples from children up to two years of age from ten Brazilian states and demonstrated that HBoV infected 15.7% of children aged 12 to 24 months. These results suggest that maternal antibodies protect children from HBoV infections, which decline at 6 months and after children are usually exposed to viral infections [30,31].

Concerning the epidemiological profile, HBoV was more frequent in children who lived in urban areas (85.4%), used water from the public network (56.2%), and lived with adequate sewage facilities (50%), although no correlation between these features and the risk of HBoV infection was shown. These features are considered as housing quality, sanitation aspects, and/or lifestyle conditions are important factors that influence health and disease in a population, including waterborne diseases, such as reported by Neves et al. [17], who investigated the presence of RVA in the same population and studied the association between public water supply, inadequate sewage facilities, and RVA infection.

Regarding economic conditions, most of the children infected by HBoV had a family income of up to three minimum wages (87.5%). Similar data were reported by Castro et al. [23] who investigated the occurrence of RVA and HBoV in immunosuppressed patients from northern Brazil and showed that 90% of patients affected by HBoV had the same monthly income. Low household incomes reduced access to adequate amounts of good-quality foods; consequently, children are prone to the development of viral infections, including diarrhea.

With respect to clinical profiles, in the present study, most of the children infected with HBoV were asymptomatic for AGE (60.4% of cases). Diarrhea, fever, and vomiting symptoms were found in 39.6%, 37.5%, and 16.7% of infected children, respectively. In a study performed in Pune, West India, with 418 fecal samples from children with AGE, it was observed that the same symptoms were more prevalent in individuals infected with HBoV; these included diarrhea (100%), fever (90%), and vomiting (58%) [32]. Such symptoms are commonly observed in HBoV-positive individuals and affected by AGE, mostly involving coinfection with other enteric viruses.

Concerning co-detection in relation to HBoV and gastroenteric viruses, HBoV was detected mostly as a single viral pathogen (83.3%) and the most prevalent coinfection was RVA+ HBoV (50%), followed by NoV+HBoV (37.5%) and HAstV+HBoV (12.5%). Distinct findings were reported by Malta et al. [24], who found HBoV monoinfection in 20.9% of strains. This difference could be attributable to the fact that the present study included individuals with and without AGE symptoms, as well as the fact that a survey was used for a few of the gastroenteric viruses.

Regarding the species analysis, it was observed that the prevalence of HBoV-1 was 45.8% (22/48), followed by other types detected (HBoV-2 and HBoV-3) in the study. This species was found as the only HBoV strain associated with viral diarrhea in children in Paraguay, with a positivity of 10.6% (37/349) [33]. Although HBoV-1 is the species most commonly related to respiratory infections in most studies, its presence in gastrointestinal infections is not null. However, there is no scientific proof whether its presence in feces is associated with infection secondary to respiratory tract infections or if it directly afflicts the gastrointestinal tract [34].

Our study had some limitations, e.g., regarding the screening of other enteric pathogens (e.g., bacteria and parasites) to elucidate the etiologic role of HBoV in acute gastroenteritis. In addition, the absence of sufficient clinical data regarding other symptoms presented by the participants to establish the association of HBoV infection and disease severity. Further studies on these aspects are warranted.

## 5. Conclusions

In conclusion, in the present study, HBoV infection is not always associated with AGE, as most HBoV cases belonged to the non-diarrheal group. However, we cannot assert that HBoV is directly associated with asymptomatic cases since there are other factors associated with viral permanence in the organism. Although HBoV is related to respiratory infections, few studies have demonstrated its relationship with gastroenteric diseases. Therefore, future studies are warranted to determine the role of HBoV in causing acute diarrhea disease

## Figures and Tables

**Figure 1 viruses-15-01024-f001:**
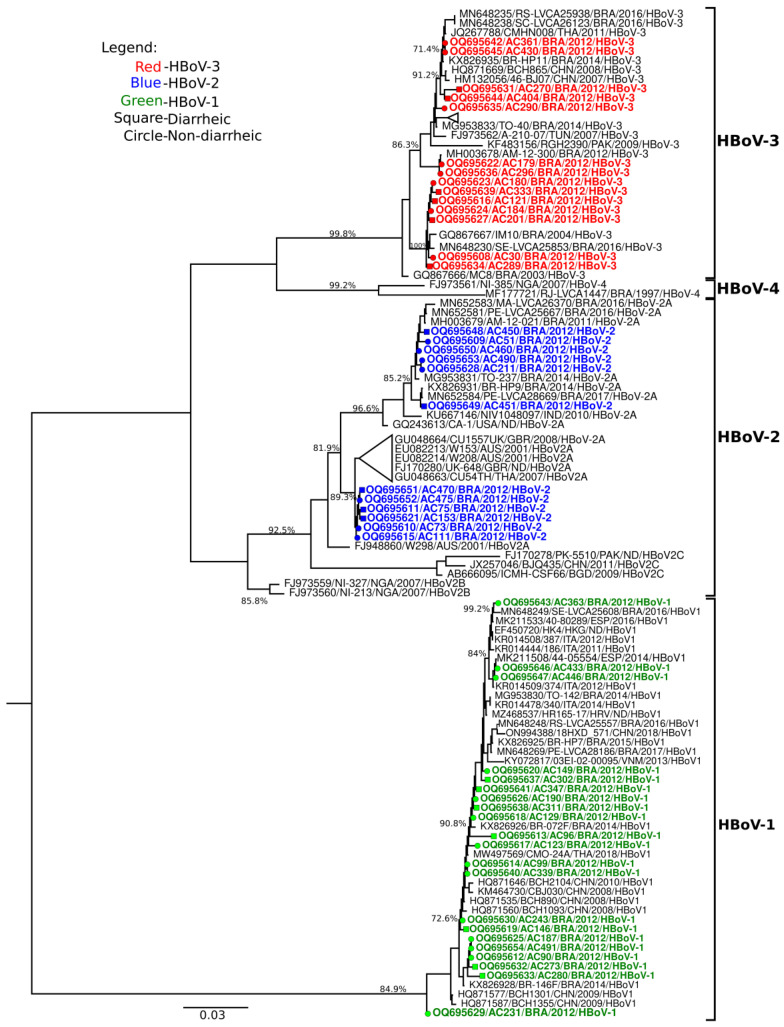
Phylogenetic tree analyses based on the VP1/VP2 gene of HBoV strains in children from Rio Branco, Acre, 2012. GenBank samples were included and accessed according to their reference numbers. The HBoV strains analyzed in this study are shown in bold and are marked in green (HBoV-1), blue (HBoV-2), and red (HBoV-3), with a square (diarrheic) or circle (non-diarrheic). The analysis was inferred using the maximum likelihood method, including the GTR+Gamma+F nucleotide substitution model.

**Table 1 viruses-15-01024-t001:** The association between epidemiological aspects and HBoV infection in children from Acre State, 2012.

Epidemiological Variables	HBoV Infection						
	Negative		Positive		Total		
	N	%	N	%	N	%	*p*
**Gender**							
Female	204	47.2	23	47.9	227	47.3	0.427
Male	228	52.8	25	52.1	253	52.7	
**Age group**							
<6 months	94	21.7	5	10.4	99	20.6	0.252
7 to 24 months	171	39.6	24	50.0	195	40.6	
25 to 60 months	165	38.2	19	39.6	184	38.4	
NI	2	0.5	0	0	2	0.4	
**Area**							
Urban	386	89.4	41	85.4	427	89.0	0.433
Rural	41	9.5	6	12.5	47	9.8	
NI	5	1.2	1	2.1	6	1.3	
**Consumer water**							
Public network supply	228	52.8	27	56.3	255	53.1	0.826
Well, river	201	46.5	21	43.7	222	46.3	
NI	3	0.7	0	0	3	0.6	
**Destination of waste**							
Sewage network	261	60.4	24	50.0	285	59.4	0.112
Septic tank	145	33.6	18	37.5	163	34.6	
Others	20	4.6	6	12.5	26	5.4	
NI	6	1.4	0	0	6	1.2	
**Income**							
**(minimum wage)**							
<1	222	51.4	25	52.1	247	51.5	0.866
2 to 3	157	36.3	17	35.4	174	36.2	
>4	34	7.9	5	10.4	39	8.1	
NI	19	4.4	1	2.1	20	4.2	
Total	432	100	48	100	480	100	

NI: No information.

**Table 2 viruses-15-01024-t002:** Clinical major symptoms observed in children infected with HBoV from Acre State, Brazil, 2012.

Clinical Features	Group						
	Diarrheic		Non-Diarrheic		Total		
	N	%	N	%	N	%	*p*
**Fever**							
No	8	42.1	22	75.9	30	62.5	0.032
Yes	11	57.9	7	24.1	18	37.5	
**Vomiting**							
No	12	63.2	28	96.6	40	83.3	0.004
Yes	7	36.8	1	3.4	8	16.7	
**Total**	19	100	29	100	48	100	

**Table 3 viruses-15-01024-t003:** Coinfection involving HBoV and other enteric viruses in children from Acre State, Brazil 2012.

ID	Group	Specie	Coinfection	Frequency
AC-121	Diarrheic	HBoV-3	*Rotavirus*	50% (4/8)
AC-280	Diarrheic	HBoV-1		
AC-289	Diarrheic	HBoV-3		
AC-404	Diarrheic	HBoV-3		
AC-270	Diarrheic	HBoV-3	*Norovirus*	37.5% (3/8)
AC-430	Non-diarrheic	HBoV-3		
AC-490	Non-diarrheic	HBoV-2		
AC-475	Non-diarrheic	HBoV-2	*Astrovirus*	12.5% (1/8)

## Data Availability

The data presented in this study are openly available in GenBank database (http://www.ncbi.nlm.nih.gov) under the access numbers OQ695608-OQ695654.

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
