# Peer review of "Epidemiologic and Clinical Characteristics of Human Bocavirus Infection in Children with or without Acute Gastroenteritis in Acre, Northern Brazil"

_viruses, 2023, doi:10.3390/v15041024_

Round 1

Reviewer 1 Report

The manuscript entitled “Epidemiologic and clinical characteristics of human bocavirus infection in children with or without acute gastroenteritis in Acre, Northern Brazil” describes the molecular epidemiology of bocavirus in Acre, Brazil during the period of one year in a study with samples from children with and without acute gastroenteritis. The English should be reviewed, the methodology used seems appropriate for the objectives of the study, however more detailed are required. My comments and suggestions made bellow are only intended to contribute to the quality manuscript.

GENERAL COMMENTS

The manuscript is well written, clear and concise and it does not need a major English language revision. It brings relevant information on bocavirus in stool samples from children with and without acute gastroenteritis symptoms; however, sample collection took place more than 10 years of the submission of the data for appreciation. My comments and suggestions made bellow are only intended to contribute to the quality manuscript.

REVIEW

ABSTRACT

The section summarizes the work well; however, I suggest re-phrasing the sentence on lines 28-30, since a percentage of samples that were positive for HBoV-2 or HBoV-3 were detected in stools from symptomatic children.

INTRODUCTION

On page1 , line 41, I suggest suppressing the text "with laboratory methods".

On page 2, line 44, I suggest replacing the word infection by symptoms.

On page 2, line 53, I suggest replacing the genotypes by species.

On page 2, line 57, add the text and without gastrointestinal symptoms.

On page 2, line 58, I suggest replacing the word prevalence by frequency.

On page 2, line 63, add the terms demographic and economic, before the text "and clinical features."

RESULTS

On page 4, lines 131-132, more detailed information on the causes of medical care/hospitalization of the non-diarrheic children should be provided, the data could be depicted in the form of a table. Also, since most HBoV-positive samples were HBoV-1, it would be important to include information on other symptoms presented by the participants, such as respiratory symptoms.

On page 4, lines 135-136 the sentence" It is noteworthy that was not register most clinical symptoms in the questionnaires" need revision.

I suggest replacing the term co-infection by co-detection, since active infection was not evaluated.

DISCUSSION

On page 7, line 184, replace the term years by old. Again, on line 184.

I suggest revising the conclusions on lines 208-210.

Authors should include in the discussion the limitations of the study.

Author Response

Dear Editors and Reviewers:
Thank you for your letters and reviewers’ comments about our manuscript entitled “Epidemiologic and clinical characteristics of human bocavirus infection in children with or without acute gastroenteritis in Acre, Northern Brazil”. The revisions were highlighted in red in the manuscript, and a line-to-line response to Reviewers’ comments was attached below. I hope this revision can make our manuscript more acceptable.
Comments and answers to reviewers
Reviewer 1:
Specific comments
ABSTRACT
The section summarizes the work well; however, I suggest re-phrasing the sentence on lines 28- 30, since a percentage of samples that were positive for HBoV-2 or HBoV-3 were detected in stools from symptomatic children.
AUTHOR RESPONSE: Thanks for your comments. We did not include the percentage of samples that were positive for HBoV-2 or HBoV-3 detected in stools from symptomatic children because HBoV-1, HBoV-2 and HBoV-3 were found in diarrheic and non-diarrheic children. Please see Page1, Lines 28.
INTRODUCTION
On page1, line 41, I suggest suppressing the text "with laboratory methods".
AUTHOR RESPONSE: Please see Page 1, line 41.
On page 2, line 44, I suggest replacing the word infection by symptoms.
AUTHOR RESPONSE: Please see Page 2, line 43.
On page 2, line 53, I suggest replacing the genotypes by species.
AUTHOR RESPONSE: Please see Page 2, line 52.
On page 2, line 57, add the text and without gastrointestinal symptoms.
AUTHOR RESPONSE: Please see Page 2, line 56.
On page 2, line 58, I suggest replacing the word prevalence by frequency.
AUTHOR RESPONSE: Please see Page 2, line 57.
On page 2, line 63, add the terms demographic and economic, before the text "and clinical features."
AUTHOR RESPONSE: Please see Page 2, line 62.
RESULTS
On page 4, lines 131-132, more detailed information on the causes of medical care/hospitalization of the non-diarrheic children should be provided, the data could be depicted in the form of a table. Also, since most HBoV-positive samples were HBoV-1, it would be important to include information on other symptoms presented by the participants, such as respiratory symptoms.
AUTHOR RESPONSE: As suggested, we included possible causes of medical care/hospitalization of the non-diarrheic children. However, we did not include detailed symptoms because most questionnaires were not filled out and it was impossible to establish the association of HBoV infection and disease severity. Please see Page 4, lines 181-183; Page 9, lines 296-299.
On page 4, lines 135-136 the sentence" It is noteworthy that was not register most clinical symptoms in the questionnaires" need revision.
AUTHOR RESPONSE: As recommended, we rewrote the paragraph. Please see Page 4, lines 181-183.
I suggest replacing the term co-infection by co-detection, since active infection was not evaluated.
AUTHOR RESPONSE: Considering the opinion, we replaced the term co-infection by co-detection in the manuscript.
DISCUSSION
On page 7, line 184, replace the term years by old. Again, on line 184. I suggest revising the conclusions on lines 208-210.
AUTHOR RESPONSE: As proposed, we included the term old. Please see Page 8, lines 233, 247.
Authors should include in the discussion the limitations of the study.
AUTHOR RESPONSE: As suggested, we included a paragraph on the limitations of the study. Please see Page 8, lines 295-299.
Reviewer 2:
As recommended, the manuscript was revised in the English language.

Reviewer 2 Report

   Dear Ms. Nola Li

   In the study  “Epidemiologic and clinical characteristics of human bocavirus infection in children with or without acute gastroenteritis in Acre, Northern Brazil” the subject of the article 2341712 is of interest in the field of Virology and Epidemiology The authors bring news of HBoV genotype and the clinical features of the infection in a study made in Northern, Brazil,  which is of little knowledge to the community.   

      Although the English text must be revised the authors presented the subject in a clear and objective way. They evaluated the clinical features and the incidence of HBoV infection in children with and without acute gastroenteritis, living in Acre, Brazil.            The introduction provided the necessary information for the reader's understanding the subject. The methodology was adequate. The results were presented in a clear and organized way. In the discussion, although the authors took into account the findings of the literature more information about the subject of the article in the other regions of Brazil should be included. Finally, the authors must review the conclusion statement.

    The article can be accepted after a revision.

Author Response

Dear Editors and Reviewers:
Thank you for your letters and reviewers’ comments about our manuscript entitled “Epidemiologic and clinical characteristics of human bocavirus infection in children with or without acute gastroenteritis in Acre, Northern Brazil”. As recommended, the manuscript was revised in the English language and the revisions were highlighted in red in the manuscript, and a line-to-line response to Reviewers’ comments was attached below. I hope this revision can make our manuscript more acceptable.
Comments and answers to reviewers
Reviewer 2:
Page 1, lines 28-31: English text must be revised.
AUTHOR RESPONSE: As suggested, we rewrote the sentence. Please see Page 1, lines 29-30.
Page 1, lines 35-37: the authors must update the references of the subject in the lines 35 to 37.
AUTHOR RESPONSE: According the recommendation, we updated the data and reference. Please see Page 1, lines 36-37.
Page 2, lines 43-63: English text must be revised.
AUTHOR RESPONSE: We revised the English text. Please see Pages 1-2, lines 42-62.
Page 7, lines 170-180: the authors must include more information about of the article in the other regions of Brazil.
AUTHOR RESPONSE: We added information on HBOV in the other regions of Brazil. Please see Page 8, lines 235-239.
Page 7, lines 180-183: English text must be revised.
AUTHOR RESPONSE: We revised the English text. Please see Page 7, lines 241-244.
Page 7, lines 191-195: English text must be revised.
AUTHOR RESPONSE: We revised the English text and we rewrote the phrase. Please see Page 7, lines 254-256.
Page 7, lines 198-199: English text must be revised.
AUTHOR RESPONSE: We revised the English text and we rewrote the phrase. Please see Page 7, lines 259-260.
Page 7, lines 205-208: But Castro et al (2019) did not compares the different social classes. Did they?
AUTHOR RESPONSE: Castro et al. (2019) compared economic conditions of immunosuppressed patients infected with HBoV.
Page 8, line 236: the conclusion of the study must be reconsidered (lines 236-237).
AUTHOR RESPONSE: We rewrote the conclusion. Please see Page 8, line 302.
